# Proteomic Analysis of *Leishmania donovani* Membrane Components Reveals the Role of Activated Protein C Kinase in Host-Parasite Interaction

**DOI:** 10.3390/pathogens10091194

**Published:** 2021-09-15

**Authors:** Sandeep Verma, Deepak Kumar Deep, Poonam Gautam, Ruchi Singh, Poonam Salotra

**Affiliations:** ICMR-National Institute of Pathology, Safdarjung Hospital Campus, New Delhi 110029, India; sanipverma6@gmail.com (S.V.); deepakhcu@gmail.com (D.K.D.); gautam.poonam@gmail.com (P.G.)

**Keywords:** host-parasite interaction, *Leishmania donovani*, membrane protein, protein-protein interaction, 2-DE

## Abstract

Visceral leishmaniasis (VL), mainly caused by the *Leishmania donovani* parasitic infection, constitutes a potentially fatal disease, for which treatment is primarily dependent on chemotherapy. The emergence of a resistant parasite towards current antileishmanial agents and increasing reports of relapses are the major concerns. Detailed research on the molecular interaction at the host-parasite interface may provide the identification of the parasite and the host-related factors operating during disease development. Genomic and proteomic studies highlighted several essential secretory and cytosolic proteins that play vital roles during *Leishmania* pathogenesis. The aim of this study was to identify membrane proteins from the *Leishmania donovani* parasite and the host macrophage that interact with each other using 2-DE/MALDI-TOF/MS. We identified membrane proteins including activated protein C kinase, peroxidoxin, small myristoylated protein 1 (SMP-1), and cytochrome C oxidase from the parasite, while identifying filamin A interacting protein 1(FILIP1) and β-actin from macrophages. We further investigated parasite replication and persistence within macrophages following the macrophage-amastigote model in the presence or absence of withaferin (WA), an inhibitor of activated C kinase. WA significantly reduced *Leishmania donovani* replication within host macrophages. This study sheds light on the important interacting proteins for parasite proliferation and virulence, and the establishment of infection within host cells, which can be targeted further to develop a strategy for chemotherapeutic intervention.

## 1. Introduction

Visceral leishmaniasis (VL) is a potentially fatal protozoan infection caused by *Leishmania donovani* on the Indian subcontinent and *L. infantum* in the Mediterranean region. In the absence of an effective vaccine, treatment of VL relies largely upon chemotherapy. Reduced efficacy and increasing relapses against available antileishmanial agents require exploring more druggable targets. Internalization of parasites and their proliferation within the host cell are of primary concern to understand the key players involved in establishing infection and parasite persistence [1]. The genome of *Leishmania* shows constitutive expression in both flagellar and aflagellar forms. Most often, mRNA abundance does not correlate with the expression level of proteins in *Leishmania* [2]. Transcriptomic studies have provided a plethora of information regarding parasitic factors associated with drug resistance, parasite fitness, and the adaptive changes during different developmental stages of *Leishmania* [3]. However, mRNA-based studies do not provide information regarding translational and post-translational modifications, which are required to understand the gene function and host pathogenesis [4]. Therefore, it becomes essential to explore the *Leishmania* proteome, which will provide a detailed understanding of gene functions and cellular reprogramming. Regulation of protein expression at developmental stages in *Leishmania* is evident at post-transcriptional and post-translational levels [5]. Recent developments in proteomic studies have opened the window for understanding disease mechanisms, identifying biomarkers for diagnosis, and identifying several vital candidates for vaccine development [6,7].

Being an obligate intracellular parasite, *Leishmania* exploits several strategies to survive and replicate within the host cell. Internalization of the parasite into the host exhibits a classical receptor-mediated process that initiates phagocytosis [8]. The host cell invasion process requires contact with the host cell plasma membrane, and the interacting proteins have been targeted in many infections [9]. Major surface molecules from both the parasite as well as the host interact with each other during the process of internalization. Upon phagocytosis, to resist degradation within the host cell to establish infection, the parasites envelope themselves within the parasitophorous vacuole, commonly known as *Leishmania* parasitophorous vacuoles (LPVs), derived from the host endocytic pathway [10,11]. Molecules from the endocytic and secretory pathways are localized and expressed on LPVs [12]. The previous study has highlighted that parasite-derived proteins are present throughout the host cell; however, their molecular contributions are poorly described to the complex host-pathogen interactions [13].

The interaction of cell membrane proteins between both the host and the pathogen is a prerequisite for the initial encounter and further penetration of the parasite to the host cell [14]. Membrane proteins have an indispensable role in host-pathogen interaction and regulatory pathways that operate within parasites and the host cells. Membrane proteins predominantly represent almost half of the potential drug targets [5].

A study on the *Leishmania* secretome highlighted several potential virulence-related proteins, and proteins associated with signal transduction, parasite survival, immuno-suppression, transport processes and antioxidant defense mechanisms [15]. Identifying such proteins involved at the host-pathogen interface will help to understand the virulence factors involved in host invasion. Such virulence factors responsible for parasite establishment may serve as new drug targets. The present study aims to identify *Leishmania* promastigote membrane proteins interacting with host macrophages using 2-dimensional gel electrophoresis (2-DE) and mass spectrometry. Such proteins may be exploited further as novel targets for chemotherapeutic intervention.

## 2. Results

Figure 1A represents the overall steps of the study design followed to identify the interacting membrane proteins. The integrity of soluble and membrane protein fractions isolated from the THP-1 macrophage and *L. donovani* parasite was verified by SDS-PAGE. The purity of membrane protein was confirmed by Western blot using an anti-sodium potassium ATPase antibody, as described in the methodology section. The band corresponding to the plasma membrane marker protein (Anti-Sodium Potassium ATPase; ~112 kDa) was observed in the THP-1 membrane protein fraction and was absent in the soluble fraction (Figure 1B).

### 2.1. Host-Parasite Interaction

#### 2.1.1. Interaction between Parasite and Macrophage Membrane Protein

The interaction between membrane proteins isolated from the parasite and macrophage (THP-1) was confirmed by ELISA using VL patient serum as the primary antibody.

There was a significant (*p* < 0.001) difference in reactivity with the VL patient serum between control (THP-1 lysate without the addition of parasite membrane) and experimental group (THP-1 membrane treated with parasite membrane protein). The OD value of the experimental group (0.964) was ~two-fold higher than the OD value of the control (0.504) (Table 1).

#### 2.1.2. Identification of Interacting Membrane Proteins by 2-DE and MS

To identify parasite membrane proteins interacting with macrophages, we labelled the parasite membrane protein with dye Cy5 and incubated with macrophages. Similarly, to identify macrophage membrane proteins that interact with *Leishmania*, the parasite was incubated with Cy5-labelled macrophage membrane protein. The cell lysate was prepared and resolved following 2-DE. The gel was then stained with colloidal CBB for picking the spot and for further characterization of the interacting proteins by mass spectrometry. Ten major spots from the parasite (Figure 2A) and six major spots from macrophage origin (Figure 2C) in the pH range of 4–7, observed on 2-DE gel using Typhoon trio scanner, were excised, trypsinized and, analyzed by mass spectrometry (MALDI-TOF/MS). MS analysis results for four spots from the parasite membrane fraction and for two spots from macrophages were obtained. The rest of the spots could not be identified due to the low abundance of protein recovered for MS analysis. Activated C kinase, peroxidoxin, small myristoylated protein 1 (SMP-1), and cytochrome c oxidase were identified from the parasite (Figure 2B), while FILIP-1 and β-actin were identified from macrophages (Figure 2D), to be involved in the host-parasite interaction. Details of the proteomic analysis are mentioned in Table 2. Further, we investigated the role of activated C kinase during parasite replication within host macrophages on the presence or absence of its inhibitor, withaferin A (WA).

#### 2.1.3. Cytotoxicity of Withaferin

We investigated cytotoxic effect of WA on the THP-1 macrophage and *L. donovani* promastigote-Ld K133. The IC_50_ value of WA for THP-1 macrophages was 4.22 ± 0.89 µM, while for promastigotes it was 1.48 ± 0.62 µM. We could observe more than 90% survival of macrophages at 1 µM concentration of WA; however, at the same concentration of WA, the percent survival of the parasite was reduced to 56%. Cytotoxicity of WA was more profound on *L. donovani* promastigotes than on THP-1 macrophages (Figure 3A,B). The further experiments were carried out with 0.1 µM and 1.0 µM concentrations of WA, as below 0.1 µM concentration, there was no significant reduction in the number of parasites when compared with the untreated control, and more than 1 µM concentration of WA was cytotoxic to host cells.

#### 2.1.4. Activated C kinase Protein Favors Parasite Proliferation within Host

*Leishmania*-activated C kinase has a well-established role in parasite viability and replication within host macrophages. In this study, in repeated experiments, we found consistent expression of activated C kinase and explored its role during host pathogenesis. To investigate the role of activated C kinase on parasite survival and persistence within macrophages, we evaluated the proliferation of *L. donovani* (*LdK133*) in macrophages in the presence or absence of WA. THP-1 and RAW264.7 macrophages were infected with the *L. donovani* standard strain AG83 and field isolate LdK133. The mean number of *L. donovani* AG83 amastigotes per THP-1 macrophage in the absence of WA (control) was 2.07 ± 0.16, while in the presence of 0.1 µM, WA was reduced to 0.99 ± 0.14. The number of amastigote per macrophages was further reduced to 0.57 ± 0.84 at 1 µM of WA. The mean number of LdK133 amastigotes per THP-1 macrophages without WA treatment was 1.96 ± 0.19, and after treatment with 0.1µM, WA and 1 µM WA were found to be 0.89 ± 0.16 and 0.60 ± 0.07, respectively (Figure 3C). The mean number of *LdAG83* amastigotes in the case of RAW264.7 macrophages in the absence of WA (control) was 3.98 ± 0.19, while in the presence of 0.1 µM and 1 µM WA, the number of amastigotes per macrophage was reduced to 1.8 ± 0.13 and 0.93 ± 0.10, respectively. The mean number of LdK133 amastigotes/RAW264.7 macrophages in the absence of WA was 3.92 ± 0.21. After treatment with 0.1 µM and 1 µM of WA, the number of amastigotes per macrophage was reduced to 1.77 ± 0.13 and 0.89 ± 0.12, respectively (Figure 3D). Significantly (*p* < 0.05) reduced proliferation of *L. donovani* in macrophages (established here using two different *Leishmania* isolates and two macrophage cell lines) in the presence of WA indicates the importance of the activated C kinase protein during the host-parasite interaction.

## 3. Discussion

In the present study, we identified a number of interacting membrane proteins involved in the *Leishmania*-macrophage interaction. Based on their biological relevance, the identified proteins belonged to different pathways/categories, including antioxidant defense (peroxidoxin), electron transport chain of mitochondria (cytchrome c oxidase), cytoskeleton, and cell motility (β-actin, SMP-1, and FILIP1), as well as a protein involved in cell proliferation and signal transduction (activated C kinase).

Study of the host-pathogen interaction is critical for understanding the disease pathogenesis. The *Leishmania*-macrophage interaction provides an excellent model for studying the proteins involved during the establishment of a successful infection, parasite survival, and persistence within the host [1,4]. Several genomic and proteomic studies have shed light on stage-specific regulation during parasite development and on membrane proteins of the parasite and host during cross-talk [16,17,18]. Proteomic studies have identified cytosolic, membrane, and secretory proteins involved in the host-parasite interaction, parasite survival, virulence, and signaling pathways, and proteins with diagnostic and vaccine potential [4,19,20,21]. Parasites circumvent host-induced oxidative and nitrosative stresses, either by manipulating their surface proteins or by modulating the host-gene expression and epigenetic modification [22,23].

In this study, we identified peroxidoxin from the parasite origin, which is a critical thiol-specific antioxidant. This protein, ubiquitous and over-expressed in the drug-resistant parasite, plays a crucial role against endogenous and host-derived oxidative and nitrosative stress [24]. Another parasite membrane protein interacting with the host macrophage was identified as SMP-1, which constitutes a major flagellar membrane protein and guides flagellar movement, as evident in the SMP-1 deleted mutant *Leishmania* promastigote [25]. Among interacting membrane proteins identified in the present study, cytochrome c oxidase, an enzyme complex located in an inner mitochondrial membrane required to meet the energy demand of the parasite to combat hostile mammalian environments/niches, was found to be important during the host-parasite interaction.

Kinases are the key molecules involved in phosphorylation of specific amino acids that play a role in different signaling cascades. They are involved in growth and proliferation and serve as important drug-target molecules [18,26,27,28]. We identified activated C kinase from the *L. donovani* membrane interacting with host macrophages. Previous proteomic studies have highlighted its role in parasite viability, progression of infection, and persistence of the parasite within the host macrophage [28,29,30]. Additionally, *Leishmania*-activated C kinase facilitates the expression of virulence-associated proteins in the mammalian host, enhancing parasite fitness during the invasion to host macrophages, and helps parasite replication in the hostile mammalian environment [29,30].

Our current finding of *Leishmania*-activated C kinase protein interacting with the macrophage membrane protein, strongly backed by previous studies, indicated this as an important molecule to explore as a potential target for chemotherapeutic intervention. Withaferin A (WA), an inhibitor of activated C kinase and other enzymes of *Leishmania*, significantly reduced the proliferation of *L. donovani* to macrophages, highlighting its role in the host-parasite interaction [31,32]. *Leishmania*-activated C kinase (LACK) facilitates the cytochrome c oxidase subunit expression to promote the fitness of *L. major*. Cytochrome c oxidase (COX) activity is favored by LACK, as evident in a study where it was shown that COX activity and oxygen consumption are reduced in LACK-deficient *L. major* [30]. Besides the parasite membrane protein, we also identified interacting membrane proteins of macrophage origin, namely β-actin and FILIP-1. Actins are conserved proteins and responsible for the integrity of the host cytoskeleton which is a requisite for parasite infection to the host [33]. Previous investigations reported that actin destabilization in the host macrophage reduced the binding of the *Leishmania* promastigote to the host, suggesting its importance during the attachment of the parasite to the host plasma membrane [33]. FILIP-1, which interacts with Filamin A, is an actin-binding protein required for cell motility. The role of Filamin A has been well studied in cancer progression and other microbial pathogenesis; however, in *Leishmania* infection biology, its role remains to be investigated [34].

In conclusion, we identified proteins involved in the host-parasite interaction that supposedly have a vital role during *Leishmania* pathogenesis and may serve as potential chemotherapeutic intervention targets.

## 4. Materials and Methods

### 4.1. Parasite Culture

*L. donovani* AG83 (MHOM/IN/83/AG83) and *L. donovani* K133 (MHOM/IN/2000/K133) parasites were cultured at 24 °C in M199 medium with 25 mM HEPES (pH7.4) supplemented with 10% heat-inactivated foetal bovine serum, 100 IU and 100 µg/mL each of penicillin G and streptomycin, respectively. Promastigotes were grown for six days to get the stationary phase parasites. Both these isolates are sensitive to various antileishmanial drugs [35,36]. LdK133, a patient-derived isolate, was used for proteomic analysis to identify the interacting membrane proteins. The validation experiments to characterize the role of LACK were conducted with both LdAG83 and LdK133.

### 4.2. Macrophage Culture

Human mononuclear cell line THP-1 and murine macrophage adherent cell line RAW 264.7 were maintained in RPMI 1640 medium, supplemented with 10% heat-inactivated fetal bovine serum, 2 mM L-glutamine, 100 U/mL penicillin, and 100 µg/mL streptomycin at 37 °C in a humidified atmosphere containing 5% CO_2_ in flat-bottom tissue-culture flasks. THP-1 cells were differentiated into macrophages by incubating them with 50 ng/mL phorbol-12-myristate-13-acetate (PMA) for 24 h. Cells were washed with 1× phosphate buffer saline (PBS) to remove non-adherent cells and PMA. Fresh medium was added, and adherent macrophages were removed after 24 h by gentle scraping [37].

### 4.3. Isolation and Quantification of Membrane Proteins

Plasma membrane proteins were isolated from *Leishmania* promastigotes and the THP-1 macrophage following standard protocol [21,38]. Briefly, parasite cells (10^10^ cells) and macrophages (3 × 10^6^) were washed thrice with ice cold 1× PBS buffer by centrifuging at 3000× *g*/10 min/4 °C. Cells were resuspended in ice-cold homogenization buffer (10 mM Tris pH 7.4, 1 mM EDTA, and 250 mM sucrose) containing a protease inhibitor cocktail (Sigma Aldrich). The suspension was lysed by sonication (25 W, 10 × 30 s) and incubated on ice for 1 h. Crude lysate was centrifuged at 15,000× *g*/30 min/4 °C, and supernatant was collected. Membrane proteins were pelleted down by ultracentrifugation of supernatant at 100,000× *g*/1 h/4 °C, and supernatant was collected as soluble fraction proteins. Membrane proteins were resuspended in buffer (40 mM Tris pH 7.4, 4% CHAPS, 2% CHAPSO). Proteins were quantified by the Bradford method.

### 4.4. Verifying Integrity and Purity of Membrane Proteins

The integrity of soluble and membrane proteins isolated was verified by SDS-PAGE. Further, the purity of membrane protein was confirmed by Western blot using an anti-sodium potassium ATPase antibody, which is a plasma membrane marker protein [39]. Proteins were separated by SDS-PAGE on 12% acrylamide gels prior to transfer onto the nitrocellulose membrane. Blot was developed using the Membrane Fraction WB Cocktail (Cat No. ab140365abcam), which contains 5 monoclonal Abs each targeting proteins located in different compartments of the cell. After primary and secondary antibodies’ incubations, the blot was developed using the ECL kit (Amersham GE Healthcare, Buckinghamshire, UK) according to the manufacturer’s protocol. The presence of the plasma membrane protein is shown by the anti-sodium potassium ATPase corresponding band (~112 kDa).

### 4.5. Evaluation of Protein Interaction by ELISA

First, we investigated the interaction of parasite membrane proteins with the macrophage membrane by enzyme-linked immunosorbent assay (ELISA Sigma-Aldrich, Saint Louis, MI, USA), as described elsewhere with modifications [38]. Briefly, the plate was coated with the THP-1 membrane protein (10 ng/well) in 0.1 M bicarbonate buffer (pH 9.0) overnight at 4 °C. The plate was blocked with 3% BSA overnight at 4 °C and incubated with the parasite membrane protein (10 ng/well) for 2 h. The plate was incubated for 2 h with the primary antibody (VL patient serum 1/100 dilution), followed by the addition of the secondary antibody (horseradish peroxidase conjugated anti-human IgG, 1/5000 dilution) and incubated for 2 h. After a subsequent wash at each step with PBST, an orthophenylenediamine (OPD) substrate with hydrogen peroxide was added, and the optical density (OD) of each well was measured at 492 nm. Wells coated with THP-1 lysate without the interaction of the parasite membrane protein was taken as a control. Wells coated with the parasite membrane protein were considered as a positive control.

### 4.6. Identification of Membrane Proteins Involved in Host-Parasite Interaction

Membrane proteins isolated from the parasite/macrophage were labeled with N-hydroxysuccinimidyl ester-derivatives of the cyanine dye (Cy5 NHS-Ester kit from Amersham, GE Healthcare, Buckinghamshire, UK). Labeling of 50 µg of membrane protein was done according to the manufacturer’s instructions. Briefly, the pH of the membrane protein was adjusted to 8.5 with 50 mM NaOH, and then Cy 5 dye (400 pico mole) was added, and kept on ice for 30 min, and then 1 µL lysine (10 mM) was added to stop the reaction. The Cy5-labeled parasite membrane protein was incubated with THP-1 macrophage for 1 h. Similarly, Cy5-labeled macrophage membrane proteins were incubated with intact parasites for 1 h.

### 4.7. Two-Dimensional Gel Electrophoresis (2 DE)

#### Iso-Electric Focusing (IEF)

2-DE of the membrane protein isolated was performed following the standard procedure [21] with modifications. Briefly, a total of 200 µg protein dissolved in the rehydration buffer (50 mM DTT, 2% of IPG-buffer, 0.002% bromophenol blue) was used for passive rehydration of the IPG-strip, pH range 3–10, 13 cm (GE Healthcare) overnight. IEF was performed on the Protean i12 isoelectric focusing (IEF) system (Bio-Rad, Hercules, CA, USA) at 20 °C for separation of proteins in the first dimension. The IEF run comprised seven steps including 250 V for 1 h, 500 V for 1 h, 1000 V for 1 h, 2000 V for 2 h, 4000 V for 2 h, 6000 V for 2 h, and 8000 V for 2.5 h, with a maximum current of a 50 µA/IPG strip. The proteins were further separated in the 2nd dimension using SDS-PAGE.

### 4.8. SDS-PAGE

The strip was sequentially equilibrated for 30 min each in an equilibration buffer (6 M urea, 0.375 M Tris–HCl, pH 8.8, 2% SDS, 20% glycerol) supplemented with 2% (*w/v*) DTT and then in an equilibration buffer with 2.5% (*w/v*) iodoacetamide. The strip was carefully loaded on the 12.5% PAGE gel, and air bubbles were removed, and sealed with 1% agarose gel. Gel was run in a SDS-electrophoresis buffer (25 mM Tris base, 192 mM glycine, 0.1% SDS) at 15 mA for 30 min and then at 30 mA until the bromophenol blue front reached the bottom of the gel. The gel was stained with Colloidal Coomassie Brilliant Blue (CBB) and scanned using the Typhoon trio using a Cy 5 setting of 670 BP 30 with red laser 633 (which transmits light between 655 nm and 685 nm and has a transmission peak centered at 670 nm). The reproducibility of the 2-D pattern was considered final when two consecutive runs produced an identical pattern with the same membrane protein fraction. The ten most intense protein spots from the parasite membrane fraction and 6 protein spots from the macrophage membrane fraction were selected for protein identification.

### 4.9. In-Gel Digestion

The in-gel digestion of protein spots of interest and purification of peptides from the gel was carried out using standard procedures described elsewhere with modifications [17]. Briefly, 2-DE protein spots (n = 10 from parasite and n = 6 from macrophage) were excised by the pipette tip, washed with HPLC grade water, and destained by a 50% acetonitrile/50 mM NH_4_HCO_3_ solution. The gel was then dehydrated with 100% acetonitrile and then rehydrated and digested with Trypsin (0.25 µg/sample) overnight at 37 °C. Peptides were extracted from the gel using 0.4% formic acid in 3% ACN twice, once using 0.4% formic acid in 50% ACN and once using 100% ACN. The extracted peptides were dried using speed vac and stored at −80 ℃ until the MALDI-MS analysis.

### 4.10. Protein Identification by MALDI-TOF/TOF

The peptide extract was reconstituted in the 10µl of 60% acetonitrile and 0.1% TFA. The peptides were mixed with the α-cyano-4-hydroxycinnamic acid matrix in a ration of 1:1 and spotted on the MALDI plate. MALDI-MS data were acquired automatically over a mass range of 0.8–3.5 kDa within a reflector ion mode at a fixed laser intensity for 1500 shots/spectrum on the 4800 MALDI-TOF/TOF Analyzer (Applied Biosystems) with the 4000 Series Explorer v3.5 software. Instrument settings were optimized to achieve optimal sensitivity at a collision energy of 1keV. The single collision condition was achieved by using air as the collision gas. The 10 most abundant MS peaks were selected for MS/MS in each MS spectrum, employing an acquisition method that excluded ions with S/N below 50. Only the strongest precursor was selected, and identical peaks detected in adjacent spots were filtered out. A MS/MS operating mode of 1 kV was used with a relative precursor mass window fixed at 250 (full-width half mass), keeping metastable suppression enabled. A total of 1250 shots with 50 shots per sub-spectrum were used for the MS/MS acquisition of selected precursors at a fixed laser intensity. The acquisition was stopped when a minimum of one hundred S/N on greater than seven peaks within the spectrum was reached after the minimum thousand shots was acquired.

To identify the peptide, peptide masses obtained from the mass spectrometric analysis were searched in the MASCOT search engine with the NCBI nr database for the *BPK282A1* strain of the *L. donovani* and *Homo sapiens* reference sequence for the identification of proteins. The detected protein threshold was a fixed confidence score of 99.9%.

The search parameters used were as follows: (a) trypsin was set as a proteolytic enzyme allowing up to one missed cleavage; (b) a precursor peptide mass error tolerance of 20 ppm was selected; (c) a fragment mass error tolerance of 0.1 Da was selected; (d) the oxidation of methionine, deamidation of asparagine and glutamine, and acetylation of protein N-termini were set as variable modifications, whereas the carbamidomethylation of cysteine was set as a fixed modification.

### 4.11. Sensitivity of Leishmania Promastigote towards Withaferin A

The sensitivity of the *Leishmania* promastigote to withaferin A (WA) was investigated by the standard resazurin assay, described elsewhere [35]. Briefly, late log phase promastigotes were plated into a 96-well culture plate (10^5^ promastigotes/well), and exposed to increasing concentrations of WA (0.0625 µM to 10 µM). Post-72-h incubation at 25 °C, 50 μL resazurin (0.0125 % (*w/v* in PBS)) were added, and plates were incubated for a further 24 h. Viability of cell was measured fluorometrically (λex 550 nm; λem 590 nm). The results were interpreted as percentage reduction in the parasite viability compared to untreated control wells. Fifty percent inhibitory concentration (IC_50_) was calculated by sigmoidal regression analysis using Microcal Origin 6.0 software (https://microcal-origin.software.informer.com/6.0/, accessed on 13 May 2021). The experiments were repeated at least twice in quadruplicates.

### 4.12. Cytotoxicity of WA on THP-1 Macrophages

Cytotoxicity of WA on THP-1was assessed using the mitochondrial-respiration-dependent 3-(4,5-dimethylthiazol-2-yl)-2,5-diphenyltetrazolium (MTT) reduction method described elsewhere [40] with minor modification. Briefly, THP-1 cells were seeded into 96 well tissue culture plate at a confluence of 30,000 cells per well and incubated with 50 ng/mL of PMA for 48 h at 37 °C and 5% CO_2._ After differentiation THP-1 cells were treated with different concentration of WA (0.0625 µM to 10 µM) for 48 h. Cells were washed with PBS and incubated with 1 mg/mL MTT in PBS for 2 h at 37 °C in 5% CO_2_, followed by DMSO treatment. Absorbance of each well was then read at 540 nm using a microplate reader (Tecan 200). The optical density of formazan formed in control cells (without treatment with WA) was taken as 100% viability.

### 4.13. Proliferation of L. donovani Parasite within Host Macrophage in Presence and Absence of Activated C kinase Inhibitor (Withaferin A)

Proliferation of the *L. donovani* parasite was investigated by infecting THP-1 and RAW 264.7 murine macrophages with *L. donovani* promastigote using standard protocols with modifications [37]. These differentiated macrophages were infected with parasites at a 1/10 macrophage/parasite ratio in a humidified atmosphere at 37 °C/5% CO_2_ overnight with or without WA (0.1 µM and 1 µM). After 24 h, slides were air-dried, fixed in absolute methanol for 5 min, and stained with Diff–Quik solutions. To calculate the number of amastigotes/cell, 100 macrophages were examined under oil immersion light a microscopy at 1000× magnification.

## Figures and Tables

**Figure 1 pathogens-10-01194-f001:**
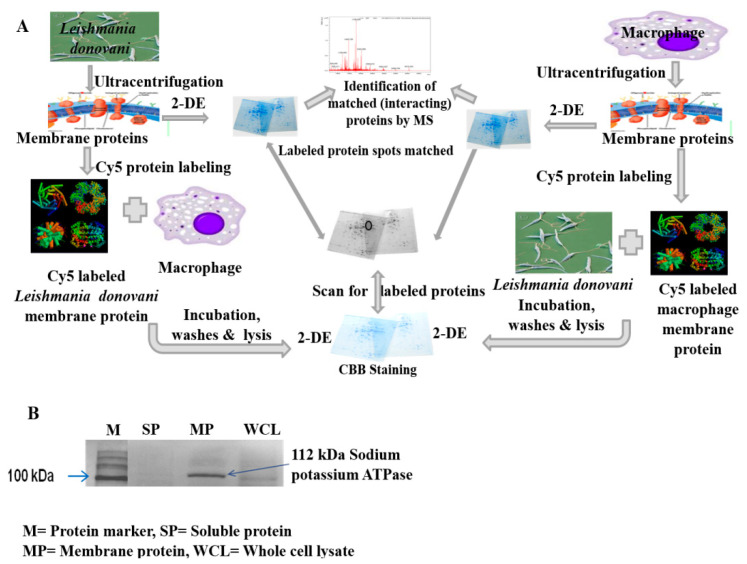
(**A**) Schematic representation of the experimental design to identify the interacting membrane proteins. Membrane proteins isolated from *L. donovani* parasite and THP-1 macrophages were labelled with Cy5 dye, resolved by 2-DE, and visualized by staining with CBB. Simultaneously, labelled membrane proteins from *L. donovani* and THP-1 macrophage were incubated with intact macrophages and *L. donovani* parasites, respectively. After incubation, cells along with attached membrane proteins were lysed and resolved by 2-DE, and images were scanned. The scanned gel image was matched with CBB-stained gel. Matched spots were excised and identified by mass spectrometry. (**B**) Western blot analysis showing enrichment of sodium potassium ATPase, a plasma membrane marker protein, in membrane fraction isolated from THP-1 macrophages. Whole cell lysate, membrane proteins and soluble proteins were resolved using SDS-PAGE and electrotransferred on PVDF membrane. Blot was probed with Membrane Fraction WB Cocktail ab. Purity of membrane protein was confirmed by plasma membrane marker sodium potassium ATPase. Membrane protein fraction showed ~112 kDa corresponding band of Plasma membrane marker protein sodium-potassium ATPase.

**Figure 2 pathogens-10-01194-f002:**
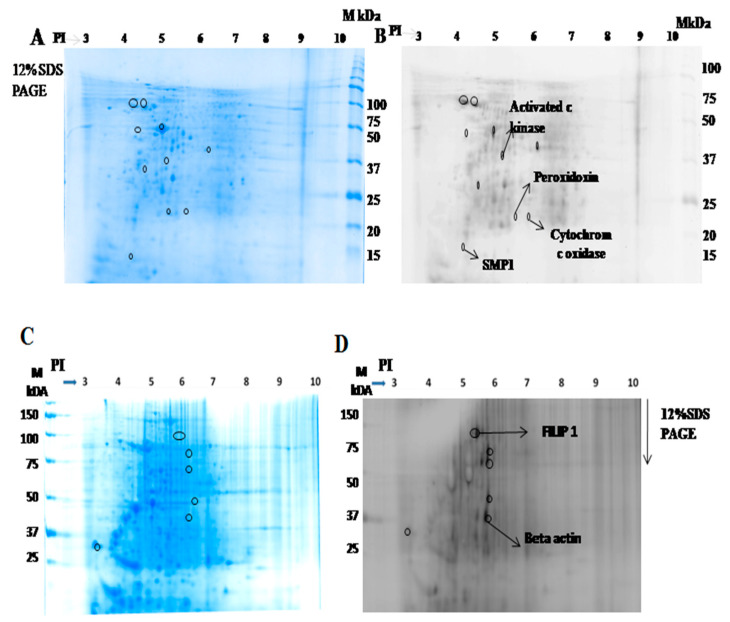
2D map of interacting membrane proteins. (**A**) Membrane proteins from *L. donovani* (100 µg) were loaded onto IPG strip pI 3–10, followed by SDS PAGE (12%), and stained with Colloidal coomassie to visualize the proteins. (**B**) Membrane proteins of *L. donovani* interacting with host macrophage proteins. Equal concentrations of Cy5-labeled membrane proteins of parasite along with THP-1 macrophage cell lysate were loaded onto IPG strip pI 3–10, followed by SDS PAGE (12%). (**C**) THP-1 macrophage membrane proteins (100 µg) were loaded onto IPG strip pI 3–10, followed by SDS PAGE (12%), and stained with Colloidal coomassie to visualize the proteins (**D**) Similar concentrations of Cy5-labeled membrane proteins of macrophages mixed along with *L. donovani* cell lysate were loaded onto IPG strip pI 3–10, followed by SDS PAGE (12%). Typhoon scanned image of 2-DE gel of membrane proteins isolated from macrophage. Circles in gel indicate the protein spots that have been processed for MALDI-TOF/MS analysis. Arrows indicate identified proteins by MS.

**Figure 3 pathogens-10-01194-f003:**
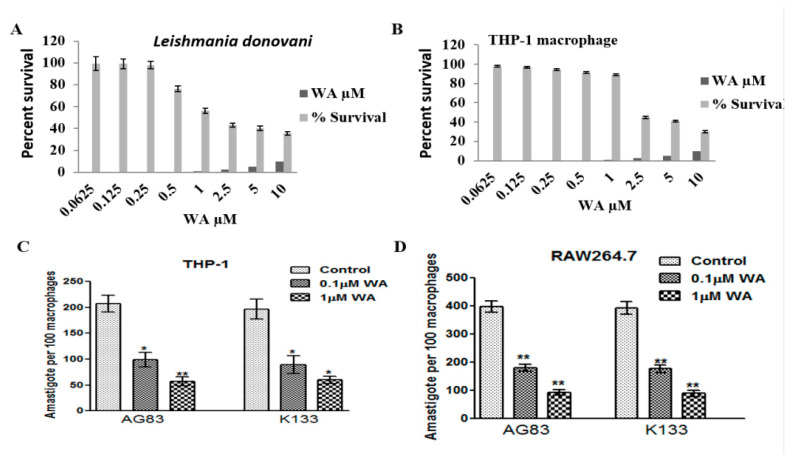
Cytotoxicity of withaferin A (WA) and its effect on proliferation of *L. donovani* in host macrophages. Cytotoxicity of WA was more pronounced on (**A**) *L. donovani* promastigote than in (**B**) THP-1 macrophages. Amastigote numbers in THP-1 (**C**) and RAW 264.7 (**D**) macrophage upon infection with *L. donovani* AG83 and K133 parasites in the presence or absence of WA, an inhibitor of activated C kinase protein. Cytotoxicity of WA in case of *Leishmania* promastigote was assessed fluorometrically by standard resazurine method. Cytotoxicity of WA in case of host macrophages was estimated using the mitochondrial –respiration –dependent 3-(4,5-dimethylthiazol-2-yl) -2,5-diphenyltetrazolium (MTT) reduction method following manufacturer’s instruction. Statistical analysis of the data was carried out using Graph Pad Prism 5 software (San Diego, CA, USA). Data are from two independent experiments each conducted in triplicate. Error bars show standard deviation. Asterisks shows level of significance (* *p* < 0.05; ** *p*< 0.001).

**Table 1 pathogens-10-01194-t001:** Confirmation of interaction between parasite and macrophage proteins.

Plate Coating	Interaction	Primary Antibody	Mean OD ± SD
Blank (BSA only)	Parasite MP	VL patient serum	0.0
THP-1 MP	Parasite MP	VL patient serum	0.96375 ± 0.03 *
THP-1 Lysate	Parasite MP	VL patient serum	1.07755 ± 0.02 *
THP-1 Lysate	None	VL patient serum	0.50365 ± 0.02 *
Parasite MP	None	VL patient serum	1.29775 ± 0.04

Values are from two independent experiments, each conducted in triplicate. MP = Membrane protein, VL = Visceral leishmaniasis, * *p* < 0.001

**Table 2 pathogens-10-01194-t002:** Details of proteomic analysis with UniProt ID and peptide sequence.

S.No	Protein Name	UniProt ID	Peptide Sequence
1	Activated protein c kinase	E9BK16	K.INVESPINQIAFSPNR.F
2	Peroxidoxin	E9BG25	R.HSTINDLPVGR.N
3	Cytochrome c oxidase	E9AGF4	R.WNLNTELHPADR.A
4	SMP 1	A4HYX4	K.MDALPLSEEYR.Q
5	FILIP 1	Q7Z7B0	K.SSELSCSVDLLK.K
6	Beta actin	Q53G99	K.DLYANTVLSGGTTMYPGIADR.M

## Data Availability

All the data is available in the manuscript.

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
