# Peer review of "Proteomic Analysis of Leishmania donovani Membrane Components Reveals the Role of Activated Protein C Kinase in Host-Parasite Interaction"

_pathogens, 2021, doi:10.3390/pathogens10091194_

Round 1

Reviewer 1 Report

Sandeep Verma and colleagues present a detailed work on Leishmania donovani-host membrane protein interaction, being able to identify several key proteins involved in the disease development. L. donovani constitutes the aetiological agent of visceral leishmaniasis (VL) on the Indian subcontinent. This disease has a significant burden of human populations and constitutes a major public health problem. The present work constitutes a valid and significant contribution to the identification of the new therapeutical targets and improve treatments.

The manuscript is well written, and the methodology presented is adequate and correct. However, there are several issues that need to be addressed.

General comments:

Line 11-12: This sentence may misleading the reader. I suggest to the authors to re-organize the sentence for example as follows: “Visceral leishmaniasis (VL), mainly caused by Leishmania donovani parasitic infection, constitutes a potentially fatal disease, which treatment is primarily dependent on chemotherapy.”

Line 13-14: Please rephrase “Investigations are warranted on the molecular interaction at the host-parasite interface to identify the parasite and host related factors operating during disease development.” by for example “ Detailed research on molecular interaction at the host-parasite interface may provide the identification of parasite and host related factors, with therapeutic potential, operating during disease development”.

Line 27: please consider the alteration of “host-parasite” to “host-parasite interaction” and “Leishmania” to “Leishmania donovani

Line 30-31: This sentence can cause some confusion on the reader as VL can also be caused by L. infantum, mainly depending on the geographic location. I suggest to the authors to include the geographic reference of the Indian subcontinent, were VL is caused by L. donovani (almost) exclusively. The authors may consider to include data from the disease burden, highlighting the data available on cases of disease relapse due to therapeutic resistance/failure.

Lines 40 and 42: Please see that bibliographic references are out of format.

Line 42: Please replace “is” by “are”.

Line 53-54: The authors may consider to develop the idea further, by addressing the role of parasite-host protein interaction as the basis of parasite internalization by macrophages, as Leishmania is completed dependent of phagocytic mechanisms of cells to be internalized.

Line 67-79: This section constitutes more a description of the experimental design followed and could be better articulated with figure 1A. The main result is the purification of protein membranes verified by the plasma membrane marker.

Line 79: “proteinfraction” please verify the need for a spacer between words.

Line 81: Please consider the rephrasing of the figure title as follow “Schematic representation of the experimental design to identify the interacting membrane proteins” and add a brief description of the experimental design followed on figure 1A description.

Line 97: Please add the information of statistical significant values to the table in the mean OD+/-SD column; Please add below the table the full word for “MP” and “VL”, as well as the number of independent repetitions of the assay represented in the table.

Line 131: The authors may consider add more information on why it is interesting to study the activated C kinase protein, from all the protein identified in the study.

Line 133: The authors may consider reformulating the figure 3A and 3B, presenting survival by log concentration of WA concentration, as the calculation of IC50 is derived from this graphical representation. The authors also, should identify what isolate of L. donovani is used to create figure 3A and a brief justification on the choosing of the concentration of WA to be tested (1 µM and 0.1 µM).

Line 140: The method used to analyse the cytotoxicity should be included in the figure caption.

Line 142: It is not clear why the authors used two different L. donovani isolates (AG83 and K133). Why use two? Do they have different drug susceptibility profile? Are they from different hosts? Which one was previously used to identify the proteins? A brief explanation is given on line 150, however these data should be more clear provided in order to elucidate the massage to the reader.    

Line 145: Please indicate the number of independent experiments represented in the figure.

Line 148: Please replace “L. donovani to macrophages” by “ L. donovani in macrophages"

Line 142 vs line 150 and 156 vs 225: Please uniformize the reference of the L. donovani isolates as they appear different: “K133” vs “LdK133” and “AG83” vs “LdAG83” throughout the manuscript.

Line 183-185: As this information is of great interest, I wonder if the authors have any information on the drug susceptibility profile of L. donovani isolates used for the present study.

Author Response

Comment 1. Line 11-12: This sentence may misleading the reader. I suggest to the authors to re-organize the sentence for example as follows: “Visceral leishmaniasis (VL), mainly caused by Leishmania donovani parasitic infection, constitutes a potentially fatal disease, which treatment is primarily dependent on chemotherapy.”

Response: As suggested by the reviewer, now the sentence has been reorganized in the text file under track change mode.

Comment 2. Line 13-14: Please rephrase “Investigations are warranted on the molecular interaction at the host-parasite interface to identify the parasite and host-related factors operating during disease development.” by for example “ Detailed research on molecular interaction at the host-parasite interface may provide the identification of parasite and host-related factors, with therapeutic potential, operating during disease development”.

Response: The sentence has been rephrased as suggested.

Comment 3. Line 27: please consider the alteration of “host-parasite” to “host-parasite interaction” and “Leishmania” to “Leishmania donovani

Response: Incorporated the suggested changes.

Comment 4. Line 30-31: This sentence can cause some confusion on the reader as VL can also be caused by L. infantum, mainly depending on the geographic location. I suggest to the authors to include the geographic reference of the Indian subcontinent, where VL is caused by L. donovani (almost) exclusively. The authors may consider including data from the disease burden, highlighting the data available on cases of disease relapse due to therapeutic resistance/failure.

Response: The sentence is modified and now read as  “Visceral leishmaniasis (VL) is a potentially fatal protozoan infection caused by Leishmania donovani in the Indian subcontinent and L. Infantum in the Mediterranean region.”

We considered the reviewer’s suggestion and realize that as per the current organization of the paragraph, the mention of disease burden and relapse rates will look out of place, and hence not added this information.

Comment 5. Lines 40 and 42: Please see that bibliographic references are out of format.

Response: Correction done.

Comment  6. Line 42: Please replace “is” by “are”.

Response: Corrected now.

Comment 7. Line 53-54: The authors may consider to develop the idea further, by addressing the role of parasite-host protein interaction as the basis of parasite internalization by macrophages, as Leishmania is completed dependent on phagocytic mechanisms of cells to be internalized.

Response: Thank you for the valuable suggestion. Now a piece of further information has been added in the introduction. Line 50 -60.

Comment 8. Line 67-79: This section constitutes more a description of the experimental design followed and could be better articulated with figure 1A. The main result is the purification of protein membranes verified by the plasma membrane marker.

Response: Now the paragraph (line 67-) starts with Figure 1A represents the…..The method part has been shifted in the figure caption

Comment 9. Line 79: “proteinfraction” please verify the need for a spacer between words.

Response: Correction made.

Comment 10.Line 81: Please consider the rephrasing of the figure title as follow “Schematic representation of the experimental design to identify the interacting membrane proteins” and add a brief description of the experimental design followed on figure 1A description.

Response: Correction made in the figure title as suggested. A brief description of experimental design shifted from the first paragraph of results to Figure 1A description.

Comment 11. Line 97: Please add the information of statistical significant values to the table in the mean OD+/-SD column; Please add below the table the full word for “MP” and “VL”, as well as the number of independent repetitions of the assay represented in the table.

Response: p-value is now added to the table. Full word for MP and VL is now added. Now, the number of independent experiments has been mentioned in table footnote.

Comment 12. Line 131: The authors may consider adding more information on why it is interesting to study the activated C kinase protein, from all the protein identified in the study.

Response: Added the information as suggested. In section 2.1.4

Comment 13. Line 133: The authors may consider reformulating the figure 3A and 3B, presenting survival by log concentration of WA concentration, as the calculation of IC50 is derived from this graphical representation. The authors also should identify what isolate of L. donovani is used to create figure 3A and a brief justification on the choosing of the concentration of WA to be tested (1 µM and 0.1 µM).

Response: LdK133 was used for the percent survival (Figure 3A) experiment. The information is now added in section 2.1.3. Further, the following justification is also incorporated.

“The further experiments were carried out with 0.1 µM and 1.0 µM concentration of WA as below 0.1 µM concentration there was no significant reduction in the number of parasites when compared with untreated control and more than 1 µM concentration of  WA was cytotoxic to host cells.”

Comment 14. Line 140: The method used to analyse the cytotoxicity should be included in the figure caption.

Response: Methods used to analyse the cytotoxicity are now included in the figure legend.

Comment 15.Line 142: It is not clear why the authors used two different L. donovani isolates (AG83 and K133). Why use two? Do they have different drug susceptibility profile? Are they from different hosts? Which one was previously used to identify the proteins? A brief explanation is given on line 150, however, these data should be more clear provided in order to elucidate the massage to the reader.   

Response:

LdK133 isolate was used to identify the proteins, and the role of selected activated c kinase was established and validated using two different isolates of L. donovani to strengthen our findings.  Both are clinical isolates derived from VL patient. Both AG83 and K133 are sensitive to sodium antimony gluconate (SAG) and other antileishmanial drugs like miltefosine and amphotericin B. We got consistent results for both the isolates and established the significance of activated C kinase in host-parasite interaction.

This information has been added in the methods section and also a brief line on the use of two isolates is added in the results section 2.1.4.

Comment 16. Line 145: Please indicate the number of independent experiments represented in the figure.

Response: The number of independent experiment is now mentioned in the figure legend.

Comment 17. Line 148: Please replace “L. donovani to macrophages” by “ L. donovani in macrophages"

Response: Correction done as suggested.

Comment 18. Line 142 vs line 150 and 156 vs 225: Please uniformize the reference of the L. donovani isolates as they appear different: “K133” vs “LdK133” and “AG83” vs “LdAG83” throughout the manuscript.

Response: Now corrected.

Comment 19. Line 183-185: As this information is of great interest, I wonder if the authors have any information on the drug susceptibility profile of L. donovani isolates used for the present study.

Response: The drug susceptibility profile of LdAG83 and LdK133 has been published in our earlier publications (Kumar et al 2009). Both LdAG83 and LdK133 are sensitive to antileishmanial drugs like sodium antimony gluconate (SAG), miltefosine (MIL) and amphotericin B (AmB).

This information is added in the methods section 4.1.  along with the reference.

Reviewer 2 Report

The aim of the authors in this work is to find key proteins involved
in Leishmania donovani-host cell interactions. It is urgent to find new
therapeutic targets to face the current problems of pharmacological
treatment (toxicity, drug resistance). The authors identify some
relevant proteins and highlight the importance of activated protein
C kinase in parasite survival and proliferation within the host cell. The experimental design is appropriate and the results are properly
ordered and support the conclusions. The methodology is adequate and
useful for similar experiments. However, revision is needed
to correct some writing errors and improve clarity.

Mistyping errors in lines 42, 79, 141, 210

In table 1, there is no description for * and ∞.

In figures 3A and 3B there is no reason to include two serial data (WA uM and % Survival). Only % Survival (without the legend) is enough to avoid misunderstandings. 

Author Response

Reviewer 2 general comments

Comment1. Mistyping errors in lines 42, 79, 141, 210

Response: Corrected.

Comment 2. In table 1, there is no description for * and ∞.

Response: Description is now added.

Comment3. CIn figures 3A and 3B there is no reason to include two serial data (WA uM and % Survival). Only % Survival (without the legend) is enough to avoid misunderstandings. 

Response: Figure 3A corresponds to the cytotoxicity of WA on L. donovani parasite whereas Figure 3B explains the cytotoxicity of WA on host macrophage (THP-1), respectively.

Round 2

Reviewer 1 Report

The authors have addressed the majority of the previously raised issues and improved manuscripts overall quality. However, there are some issues that still need to be addressed before manuscript final acceptance.

As major issue, the figure 3 is missing from the manuscript. Please correct. 

There is also, the need to perform several minor corrections of English language and style. I recommend an attentive revision of manuscript. For example: 

Line 16: “Genomic” – no need for capital letter

Line 14 – please remove the extra space between words 

Line 33: “ L. Infantum” – "infantum" does not use the capital letter

Line 150 and 176 – lack of space between words

Line 165 – "florimterically" – please correct to "fluorometrically"

Line 355- 370 - please verify the text formatting style as it is different from the rest of the text.

Author Response

  • A major issue, the figure 3 is missing from the manuscript. Please correct. 

          Response: Fig 3 is included now.

There is also, the need to perform several minor corrections of English language and style. I recommend an attentive revision of manuscript. For example: 

Line 16: “Genomic” – no need for capital letter

Response: It is a new sentence,appropriate correction was done

Line 14 – please remove the extra space between words 

Response: Corrected.

Line 33: “ L. Infantum” – "infantum" does not use the capital letter.

Response: Corrected.

Line 150 and 176 – lack of space between words

Response: Corrected.

Line 165 – "florimterically" – please correct to "fluorometrically"

Response: Corrected.

Line 355- 370 - please verify the text formatting style as it is different from the rest of the text.

Response: Formatted.

The whole manuscript has been checked for formatting errors, and necessary corrections were done.